# The Potential Application of Endophytes in Management of Stress from Drought and Salinity in Crop Plants

**DOI:** 10.3390/microorganisms9081729

**Published:** 2021-08-13

**Authors:** Hariom Verma, Dharmendra Kumar, Vinod Kumar, Madhuree Kumari, Sandeep Kumar Singh, Vijay Kumar Sharma, Samir Droby, Gustavo Santoyo, James F. White, Ajay Kumar

**Affiliations:** 1Department of Botany, B.R.D. Government Degree College Duddhi, Sonbhadra 231216, India; vermahariom87bhu@gmail.com; 2Centre for Advanced Study in Botany, Banaras Hindu University, Varanasi 221005, India; dharambhu@gmail.com (D.K.); sandeepksingh015@gmail.com (S.K.S.); 3Department of Chemistry, BSA College, Mathura 281001, India; vchem20@gmail.com; 4Indian Institute of Science, Bengaluru 560012, India; madhuree88@gmail.com; 5Volcani Center, ARO, Department of Postharvest Science, Bet Dagan 50250, Israel; vjsharma@outlook.in (V.K.S.); samird@volcani.agri.gov.il (S.D.); 6Instituto de Investigaciones Químico Biológicas, Universidad Michoacana de San Nicolás de Hidalgo, Morelia 58030, Mexico; gustavo.santoyo@umich.mx; 7Department of Plant Biology, Rutgers University, New Brunswick, NJ 08901, USA

**Keywords:** abiotic stress, endophytes, reactive oxygen species (ROS), stress genes, plant defense system

## Abstract

Endophytic microorganisms present inside the host plant play an essential role in host fitness, nutrient supply and stress tolerance. Endophytes are often used in sustainable agriculture as biofertilizers, biopesticides and as inoculants to mitigate abiotic stresses including salinity, drought, cold and pH variation in the soil. In changing climatic conditions, abiotic stresses create global challenges to achieve optimum crop yields in agricultural production. Plants experience stress conditions that involve endogenous boosting of their immune system or the overexpression of their defensive redox regulatory systems with increased reactive oxygen species (ROS). However, rising stress factors overwhelm the natural redox protection systems of plants, which leads to massive internal oxidative damage and death. Endophytes are an integral internal partner of hosts and have been shown to mitigate abiotic stresses via modulating local or systemic mechanisms and producing antioxidants to counteract ROS in plants. Advancements in omics and other technologies have been made, but potential application of endophytes remains largely unrealized. In this review article, we will discuss the diversity, population and interaction of endophytes with crop plants as well as potential applications in abiotic stress management.

## 1. Introduction

Global agricultural productivity is largely influenced by various abiotic factors including drought, salinity, cold, heat and variations in soil pH that hamper optimum agricultural yields. Changing climatic conditions and rising anthropogenic activity of growing populations accelerates the challenges of abiotic stresses [1]. However, uncertainty of climatic condition, irregularity in rainfall, heat waves and rise in the global temperatures directly affect the optimum growth and yield of crop plants because of their direct effect on reducing water availability, decreasing photosynthetic rates and creating drought conditions [2,3]. Under severe drought the level of water in the soil falls, while the salt content is increased, leading to osmotic stress and higher concentrations of salinity result in ionic toxicity and osmotic stress in roots [4]. Osmotic stress helps plants to absorb water but in saline soil, the osmotic pressure of soil solution surpasses the plant osmotic pressure and thereby reduces the uptake of water into the plant. In such circumstances, not water but ions such as Na^+^ and Cl^−^ move into the plant [5]. In the current era, drought and salinity are the two most severe abiotic stress factors that affect growth and productivity of plants globally [4,6].

Drought is one of the most severe and emerging abiotic stresses that affect growth and productivity of plants via affecting several physiological and metabolic processes in crop plants [7]. Drought stress has drastic impacts on root physiology, leaf structure, nutrient uptake, photosynthetic activity and seedling germination, resulting in overall decreased growth of agricultural crops [8,9]. However, effects of drought on the plant system depends on the intensity and duration of exposure. Under short term drought, plants systems increase water use efficiency by reducing stomatal aperture and transpiration rate [10], while long term exposure of drought disrupts chloroplasts and starch granules, which directly affect photochemical activities and decrease transpiration rate of the plant [11,12].

Similarly, salinity is another challenging abiotic stress factor that severely affects physiological and metabolic processes of plants through reduce seedling growth, decreased photosynthetic activity, water stress, ion toxicity and decreased rates of protein synthesis and lipid metabolism [13,14]. Currently it has been estimated that approx. 20% of the total cultivable land faces saline stress globally and this will reach to 30% by 2050 [15]. Additionally, low rainfall and high temperatures both play crucial roles in increasing soil salinity, mainly in the arid and semi-arid regions of the world [16]. The severity of salinity on plant cells depends upon salt concentration and exposure time [17]. The onset of salinity can be seen as water stress that results in reduced leaf expansion, which further turns into complete inhibition of cell division and stomatal closure, while long-time exposure leads to premature leaf senescence resulting in decreased photosynthetic activity and ultimately death of crop plants [18,19].

Under stress, plants evoke a series of reactions in terms of signal transductions, stress responsive genes, activation or inactivation of functional proteins and responses in particular cell organelles, mainly chloroplasts, mitochondria and peroxisomes, to develop stress tolerance [20]. Plant systems also elevate their molecular behavior under stress conditions by secreting stress hormones and ROS, which functionally regulate cellular physiology to maintain normal functioning of plants [21,22].

However, to mitigate the challenges of abiotic stress and their impact on growth yield and productivity of plants, utilization of beneficial microbial strains is the most feasible, reliable and sustainable option [23]. It is known that plant microbial communities play an integral role in maintaining or enhancing growth and fitness of plants under various biotic and abiotic stress conditions [24]. In this review paper, we summarize research regarding endophytic microbial strains that are an integral part of a beneficial and sustainable approach in control of abiotic stresses including drought and salinity stresses.

## 2. Microbial Endophytes

Plants are host to microbial communities that include bacteria, archaea and fungi as epiphytes or endophytes. Even though, plant compartments, including phyllosphere (leaf surfaces), carpophore (fruit surfaces) and rhizosphere (root surfaces) harbor large numbers of microbes some of the microbes reside as endophytes inside the plant tissues without showing any external or apparent signs of infection [25,26]. De Bary [27] first introduced the term “endophyte” for microbes living inside plant tissue without causing any signs of infection. Petrini [28] defined endophytes as microorganisms that reside for some part of their life cycles inside host tissues. The term endophyte was first used for fungal species and later for bacterial species within plant tissues [25,29], With an advancement of omics and similar techniques, exploration of endophytic microbial communities has advanced and it has been recognized that all plants have endophytic microbes at all stages of their life cycles [30]. The dominant phyla of prokaryotic endophytes reported in the main databases (96%) include those 16S rRNA gene sequences belonging to Proteobacteria (54%), Actinobacteria (20%), Firmicutes (16%) and Bacteroidetes (6%). Members of the genus *Pseudomonas*, *Enterobacter*, *Pantoea*, *Stenotrophomonas*, *Acinetobacter* and *Serratia* are part of the main *Gammaproteobacteria* found as endophytes of various plant host species, and therefore, knowledge of them is deeper with respect to other less explored bacterial endophytes [25]. In the case eukaryotic endophytes, Hardoim and colleagues [25] report that there are in databases internal transcribed spacer (ITS) sequences assigned to the main phyla *Glomeromycota* (40%), *Ascomycota* (31%), *Basidiomycota* (20%), unidentified phyla (8%) and *Zygomycota* (0.1%). The phylum *Glomeromycota* only includes arbuscular mycorrhizal fungi (AMF), whose species have been reported as restorers of degraded ecosystems and facilitators of plant growth under diverse stress conditions [8].

Endophytes provide support in acclimatizing crop plants under abiotic stress conditions, growth promotion and management of phytopathogens, and they help in activating stress responsive/induced genes of plants that are not usually activated under stress conditions. An overview of endophyte-mediated mechanisms for drought and salinity stress management in crop plants is provided in Figure 1.

## 3. Entry and Colonization of Plants by Microbial Endophytes

During initial colonization to the host surface, endophytic microbes are confronted with immune response of the host. This may be overcome depending on the endophytic microbial strains and the particular host colonized. Successful colonization of the endophytic microbial strains is mediated by a series of reactions that are completed in several steps and regulated by genetic, metabolic or growth regulator factors [31]. In the initial step of colonization, microbes are attracted to the host surface (e.g., roots), which is facilitated through chemical exudates, including polysaccharides, amino acids, flavonoids, organic acids, etc. that act as chemo-attractants and nutrients for the microbes. Microbes move towards the host surface with the help of flagella, pilli or fimbri appendages [29], and may secrete biochemical compounds such as exopolysaccharides (EPS), lipopolysaccharides or biofilms that help in attachment of microbes to the plant surface to begin colonization [32].

The exopolysaccharides secreted by the bacterial cell facilitate endophyte colonization through attachment of bacterial cell to the host surface. Meneses et al. [33] reported, EPS secreted by *Gluconacetobacter diazotrophicus* Pal5 play an essential role in attachment and colonization of endophytic strain to the root surface of rice. Moreover, bacterial endophyte enters through the cell wall of the host via secreting cell wall degrading enzymes such as cellulases, endoglucanases and pectinases, that facilitate entry and colonization in the host tissue [33]. Reinhold-Hurek et al. [34] reported about endoglucanase mutant strain *Azoarcus* sp. BH72, having lower entry frequency of mutant strain in comparison to the wild strain. Additionally, the mutant strain is not able to spread at the aerial plant parts. However, colonization of endophytic fungi might be initiated through attachment of strain to the host surface and forming appressorium-like structures. Subsequently, fungal endophyte colonizes and migrates to the internal tissue via penetrating outer surface of the host plant [35]. Even though, cell wall remains intact during early colonization of *Trichoderma* to the tomato roots as observed in the microscopic study. However, in certain cases higher number of extracellular enzymes was reported in the host tissue during endophytic fungal colonization [36].

However, successful colonization of endophytes into plants involves compatible plant-microbe interactions, signaling molecules between microbes and host tissue [32,37] and depends upon various factor such as nature of microbe, host genotypes, plant exudates, nutrient availability, stress factors, as well as the surrounding environment [25]. The nature of a microbe is specific for the host plant or related group of hosts and may be mutualistic, neutral or pathogenic. The colonization patterns and efficacy of microbial strains are unique depending on the plant and microbe. For instance, pathogenic strains secrete higher amounts of cell wall degrading enzymes in comparison to symbiotic endophytes, and the entry of pathogenic strains causes increased hypersensitivity in host plants; however, symbiotic strains do not show this effect in host plants [29]. Using histochemical analysis, Chang et al. [38] proposed that secretion of the growth regulator ethylene by root cell intracellular endophytes was a first key communication in the microbe to plant interaction that triggered host root hairs to grow and release nutrients (exudates) and superoxide. To protect themselves from superoxide, endophytes produced antioxidants, including nitric oxide, to denature superoxide. These two chemical interactions between endophytes and plant cells represent key nutrient exchanges of carbon and nitrogen between the symbionts. Endophytes have also been shown to produce phytohormones, and these too may play roles in the interaction between microbe and plant [25]. Once in host tissues, endophytes move within plants via the conductive tissues, xylem and phloem [39,40]. Microbes enter into host tissues at plant meristems (root and shoot meristems) and through natural openings such as stomata, wounds, aerial parts of the plants, cotyledons or through root zone aerial parts of the plants [38,41]. The composition and diversity of endophytic microbes depends upon several factors including host genotype, plant age, plant organs, seasons and surrounding biotic and abiotic stress factors [42,43]. The physiology and metabolism of plants strongly depends on and is influenced by the associated microbiome under natural conditions [44]. Plant associated endophytic microbiomes regulate adaptive behavior against biotic and abiotic stress factors. In addition to these factors, climatic conditions, environmental stress, temperature, moisture content also influence the endophytic microbial diversity of the host plant [29]. Drought conditions affect root morphology of plants, leading to secretion of root exudates and changes that affect composition and chemical compounds in exudates, thus affecting diversity and abundance of microbes. Similarly, changing seasonal conditions also affect the microbial composition because of variation in the concentration of amino acids, proteins, sugar and organic acids [45].

Recent hypothesis proposed by Oono et al. [46], regarding endophyte species richness in the plant surviving under stress conditions. The lower nutrients and higher concentration of toxic compounds can limit the growth of fungi that may increase their diversity and predict species richness of endophyte, due to suppression of otherwise dominating species. Even though, differences in the endophytic diversity depends upon several factors such as host specificity, microclimatic conditions, seasonality. For instance, strong dry seasons can act as physiological filter for horizontally transmitted fungi, that present outside of leaves for parts of their life cycle that potentially led to lower richness of the local species pool of endophytes [47,48].

## 4. Reactive Oxygen Species and Abiotic Stress Factors

Reactive oxygen species (ROS) may be considered endogenously produced signal molecules or regulators produced by several plant organelles, including mitochondria, chloroplast or peroxisomes under stresses. ROS consist of a group of chemically reactive oxygen molecules such as hydrogen peroxide (H_2_O_2_), superoxide radical (O_2_•-), hydroxyl radical (OH•) and singlet oxygen (^1^O_2_) and are produced in plants under stress conditions [49,50].

Abiotic stress leads to overproduction of ROS that must be managed in a homeostatic pool; however, excess concentrations of ROS cause oxidative stress, which results in denaturation of protein structure, lipid peroxidation, nucleotide disruption and may affect plant physiology which ultimately leads to the death of plants [51].

In the plant system, mitigation of ROS excess concentrations generally leads to activation of either enzymatic or non-enzymatic antioxidant systems. Plants secrete several enzymes, including catalase (CAT), ascorbate peroxidase (APX), superoxide dismutase (SOD), glutathione reductase (GR), dehydroascorbate reductases (DHAR) and monodehydroascorbate reductases (MDHAR,); the nonenzymatic system involves quenching of ROS via synthesis of ascorbic acid (AsA), glutathione (GSH), carotenoids which quench free radicals and protect the plant cell from oxidative stress [52,53].

## 5. ROS and Signaling Molecules under Abiotic Stress

The plant system shows adaptive response under stress conditions at certain levels, via activating stress tolerance genes. However, crossing the tolerance limits, the sensor presents in the plant system, for instance gene COLD1, responsible for detecting cold stress in rice, senses the stress signal and responds [54]. Under homeostatic conditions, plants maintain a fine balance between production and quenching of ROS, while overproduction of ROS at the cellular level under stress conditions hampers the natural physiological or metabolic state of plants. ROS, however, play significant role in various functions including development via oxidizing polysaccharides in cell walls [55] or programmed cell death [56]. Moreover, the ROS produced in the various cellular compartments alter transcriptional or transcriptome levels [57]. The production, temporal and spatial distributions of ROS under different environmental conditions act as signal molecules [58]. Under stress conditions, NADPH bound by cytosolic membranes produces hydrogen peroxide (H_2_O_2_) that acts as a signaling molecule. Even though, production of moderate levels of H_2_O_2_ and O_2_ in peroxisomes act as signaling molecules [59]. In addition to ROS, several other signaling molecules such as phytohormones especially ABA, ethylene Ca^2+^, NO_2_, inositol phosphates and systemin, also serve as signaling molecules. The signaling molecules actively involved in regulating various biological functions of the plants system including modulation of gene expression, homeostasis under stress conditions [60,61].

## 6. Endophyte Mediated Drought and Salinity Stress Management

The endophytic microbiome shows mutualistic relations with the host plant in maintaining health or vigor [25,62,63]. Moreover, also essentially involved directly or indirectly in the growth and development of host plants via secreting various growth promoting attributes viz. phytohormone synthesis, nutrient acquisition and siderophore production, antibiotic phosphate solubilization, and by mitigating various biotic and abiotic conditions [64,65].

However, this has been explained in previous studies on the impacts of drought and salinity stress on the effect of growth, productivity or survivability of plants [63]. In a study Zhou et al. [66] reported improved seedling growth of *Pinus tabulaeformis* after inoculation of endophytic strain *Phoma* sp under draught condition. Wu et al. [67] reported decreased leaf area, photosynthetic pigments and photosynthetic efficiency under drought stress. Higher salinity in soil affects the survivability of plants by altering chemical, morphological and physiological processes [15].

In this context, microbial endophytes appear to be a suitable alternative for drought and salinity stress management. In the recent past, various microbial strains have been successfully utilized to increase drought tolerance. Inoculation of microbial endophytes or exogenous supply of phytohormones, significantly enhanced adaptive behavior of plants via improving photosynthetic activity, chlorophyll contents, root growth, water status, antioxidant enzymes, phytohormone signaling and nutrient uptake under drought conditions [68,69,70,71,72].

The latest published reports reinforce the utilization of endophytic strains in abiotic stress management. Naveed et al. [71] reported improved growth, water availability, as well as photosynthetic activity in maize cultivars under drought after inoculation of endophytic bacterial strains *Burkholderia phytofirmans* strain PsJN and *Enterobacter* sp. FD17. The endophytes inoculation improved seedling growth, shoot and root biomass and photochemical efficiency of PSII. Yandigeri et al. [71] demonstrated potential of endophytic bacterial strains *Streptomyces coelicolor* DE07, *S. olivaceus* DE10 and *S. geysiriensis* DE27, isolated from arid and drought affected regions, to increase tolerance of plants to intrinsic water stress and showed plant growth promotion after application to wheat seedlings. Additionally, the combined application of *S. olivaceus* DE10 + *S. geysiriensis* DE27 strains showed synergistic effects and showed improved response in terms of stress mitigation and growth promotion.

Jayakumar et al. [73] reported that several endophytic bacterial strains, including *Bacillus* sp., *Providencia* sp. and *Staphylococcus* spp., isolated from *Ananas comosus*, enhanced drought tolerance, and promoted growth as well as pathogen resistance. Similarly, Sandhya et al. [74] reported that several endophytic bacterial strains isolated from various crops in which most of the strains conferred drought tolerance up to (−1.02) matric potential also had growth promotion potential. Chen et al. [75] reported endophytic strain *Pantoea alhagi* isolated from *Alhagi sparsifolia*, after inoculation, enhanced the growth of wheat seedlings under drought conditions, additionally the endophyte-treated plant showed enhanced accumulation of soluble sugars and decreased concentrations of malondialdehyde. In the grass *Brachypodium distachyon*, drought stress was mitigated with the help of an endophytic bacterium *Bacillus subtilis* B26, which also upregulated the stress responsive genes [76]. Morsy et al. [77] reported that the endophytic fungal strains *Ampelomyces* sp. and *Penicillium* sp., isolated from stress inducing soil (drought and high salinity), enhanced drought tolerance (*Ampelomyces* sp.) and salinity tolerance (*Penicillium* sp.) in tomato. Table 1 summarizes some of the works reviewed here.

## 7. Phytohormone Modulation of Oxidative Stress Tolerance

It is well established that phytohormones play an essential role in maintaining the normal physiological and metabolic behavior of plants under stress conditions via adaptive responses [86,87]. Auxin (IAA), cytokinin (CK), gibberellin (GA); ethylene and abscisic acid (ABA) are the most common. Salicylic acid (SA), nitric oxide (NO), nitrogen dioxide (NO2), strigolactone (SL) and brassinosteroids (BR) are additional phytohormones that may also regulate plant growth under normal or stress conditions [88]. The coordinated synergistic and antagonistic effects of phytohormones essentially play an active role in stress management. The hormones auxin and cytokinin promote stomatal opening, ABA and ethylene regulation lead to stomatal closure under drought stress conditions [87].

Endophytic microorganisms produce hormones are stimulate indigenous levels of plant hormones; thus, endophytes modulate developmental or signaling processes in plants [89]. The phytohormones ethylene, IAA, GA and cytokinins are very commonly synthesized by endophytic microbial strains, which directly or indirectly modulate the growth of host plant cells and tissues [90]. The function of endophyte-produced phytohormones is very likely to stimulate plant cell growth in order to trigger release of nutrients to the endophytes [30]. There are numerous reports that show the effect of endophyte phytohormones to mitigate abiotic stress. Waqas et al. [91] reported improved macronutrient absorption in soyabean after inoculation with phytohormone secreting *Galactomyces geotrichum* endophytes. Zamioudis et al. [92] reported auxin transport potential of a strain of *Pseudomonas* that improved the architecture of the *Arabidopsis* root system. Verma et al. [90] reported auxin synthesis by strains of *Pseudomonas* sp. and *Pantoea dispersa* isolated from rice seeds, which after inoculation enhanced root and root hair growth of rice seedlings. Similarly, Shahzad et al. [93] reported that endophytic bacterial strain *Bacillus amyloliquefaciens* from rice seeds produced gibberellins (GAs) and their functional aspect of improving host physiology. The inoculation of *B. amyloliquefaciens* significantly enhanced SA production and decreased the concentration of endogenous abscisic acid and jasmonic acid in rice seedlings.

Further, in a study, Shahzad et al. [94] reported efficacy of the endophytic bacterial strain *Bacillus amyloliquefaciens*, which after inoculation significantly produced ABA with beneficial responses in the plant in mitigating salinity stress in the plant. Additionally, the rice inoculated with *B. amyloliquefaciens* significantly enhanced growth as well as enhanced levels of some essential antioxidant amino acids such as cysteine, aspartic acid, glutamic acid, phenylalanine and proline under stress conditions. Similarly, Bodhankar et al. [95] studied pre-treatment effects of maize seed with the endophytic strains *Corynebacterium hansenii* and *Bacillus subtilis*, which after inoculation improve growth and physiology of maize under drought stress. In addition, pre-treatment with *C. hansenii* improved relative water content, leaf proline and chlorophyll contents, whereas pre-treatment with *B. subtilis* enhanced fresh or dry weight of maize over the control plants under drought conditions. Rehman et al. [96] tested an endophytic *Pseudomonas* sp. strain, which after seed priming, improved growth and Zn status in the wheat. Even though, the author reported maximum yield enhancement after seed priming whereas soil and foliar application improved protein content, Zn concentration in the aleurone layer, endosperm and also in the overall grain.

In addition, ACC deaminase (1-aminocyclopropane-1-carboxylate) enzymes synthesized by endophytic microbial strains lower the ethylene levels in plants during stress conditions [97]. Jaemsaeng et al. [79] reported that the endophytic bacterial strain *Streptomyces* sp. imparted enhanced salt tolerance in rice through the action of 1-aminocyclopropane-1-carboxylate deaminase (ACCD) by converting a precursor of ethylene into ammonia and α-ketobutyrate, which consequently reduced ethylene levels in plants. Similarly, inoculation of *Nicotiana attenuata* with *Sebacina vermifera* improve fitness by altering ethylene signaling by the reduction of 1-aminocyclopropane-1-carboxylic acid (ACC) [98].

Further, in a study by Barnawal et al. [83], investigators reported improved salinity stress tolerance in *Chlorophytum borivilianum* after inoculation with endophyte *Brachybacterium paraconglomeratum* that decreased the concentration of ethylene through the deamination of ACC. In addition, improved growth and higher levels of antioxidant proline was observed in the endophyte treated plants. It is evident that ethylene is a key hormone that is impacted by endophytes that increase root growth and confer increased stress tolerance [41]. Other compounds that include antioxidant nitric oxide and nitrogen dioxide, that may also be growth promotional, may play key roles and act synergistically with ethylene to modulate plant stress [41]. Numerous papers suggest that ACC deaminase is a mechanism that contributes to increased endophyte-mediated stress tolerance, however, other evidence indicates that ACC deaminase is an incomplete or incorrect mechanism to explain endophyte-mediated stress tolerance [25,41]. Future work is needed to identify the precise microbe-plant interactions that result in endophyte-mediated stress tolerance.

## 8. Endophyte-Mediated Oxidative Stress Management

ROS generation within plants naturally occurs as plants undergo normal metabolic activities [99,100]. Under normal conditions, ROS act as signaling molecules and serve to maintain symbiosis with intracellular symbiotic bacteria [30,41,101], however, plants maintain homeostasis conditions through use of ROS scavengers, including antioxidant amino acids, enzymes and other antioxidant systems. However, the state of loss of equilibrium between generation and scavenging of ROS leads to excess ROS and oxidative damage to nucleotides, proteins, lipids and ultimately cell death [96]. Moreover, each cell compartment has specific mechanisms of ROS homeostasis or signaling depending upon the type of cell, level of stress and ROS gene network [58,102,103].

Inoculation of endophytic microbes into plants significantly mitigates the damage of oxidative stress caused by abiotic stress agents. The mechanisms used by endophytic microbes against salinity stress are similar to those used for drought stress. Moreover, colonization by endophytic microorganisms enhances plant levels of antioxidant enzyme concentrations such as (catalase (CAT), superoxide dismutase (SOD), peroxidase (POD) ascorbate peroxidase (APX) [63,104] or non-enzymatic antioxidant molecules such as AsA, GSH and carotenoids [105,106].

Endophytes induce synthesis of antioxidants to balance an array of free radicals that maintain normal cellular functioning. In addition, production of osmolytes maintain sodium-potassium ratio, which overcome the osmotic effect caused by stress factors [83]. Published reports in the recently reinforce findings of effective endophyte-modulated tolerance to ROS in plants under salinity and drought stress conditions [107,108]. Redman et al. [107] observed that endophyte inoculations significantly decreased the accumulated ROS in the plant cell by activating antioxidant enzymes. Baltruschat et al. [85] reported reduced levels of CAT, APX, GR DHAR in root tissues of barley under saline conditions. However, root colonization by *Piriformospora indica* elevated the antioxidant enzyme and ascorbic acid in the barley roots. Additionally, inoculation by *P. indica* significantly enhanced plant growth and attenuated NaCl-induced lipid peroxidation. Similarly, Zhang et al. [109] evaluated colonization potential of *Trichoderma longibrachiatum* T6 in wheat seedlings under 150 mM NaCl saline concentration. However, endophyte inoculation significantly enhanced chlorophyll content, root activity and proline accumulation in leaves. The inoculation significantly enhanced the concentration of antioxidant enzymes, mainly SOD, POD, CAT in wheat seedlings. Azad and Kaminskyj [110] reported that endophytic fungal strains *Alternaria* spp. and *Trichoderma harzianum* inoculation into tomato seedlings under salinity and drought stress conditions resulted in maintenance of photosynthetic efficiency and effectively reduced ROS accumulation. Abd-Allah et al. [78] extensively studied the inoculation impact of *Bacillus subtilis* in chickpea plants under saline conditions and observed enhanced levels of ROS scavenging antioxidant enzymes superoxide dismutase, peroxidase, catalase and glutathione reductase as well as ascorbic acid and glutathione; *B. subtilis* also enhanced plant biomass and photosynthetic pigments. Ahmad et al. [111] evaluated inoculation impact of *Trichoderma harzianum* in mustard seedlings and found that it significantly enhanced shoot and root length, and plant dry weight compared to non-inoculated plants under salinity stress conditions. Moreover, endophyte inoculation significantly enhanced the oil content and chlorophyll ‘a’, which was negatively impacted by NaCl concentration, in addition proline concentration was also enhanced, showing modulation of osmolytes and antioxidants in mustard seedlings. Therefore, exclusion or accumulation of Na^+^ concentration in the cell sap or plant cell is necessary to avoid stress [13,112]. To avoid the oxidative stress caused by Na^+^ in plants, exclusion of Na^+^ from the leaf surface is the most common phenomenon and reported by various authors from cereal crops studies [113]. However, failure of Na^+^ exclusion affects a plant’s morphology and causes premature death of older leaves, and the effect of toxicity varies with plant species and duration of exposure [13].

In a pot experiment root colonization by *Pseudomonas pseudoalcaligenes* of the model plant *Arabidopsis* improved growth under salinity stress and the possible reason for that tolerance was modulation in the expression levels Na+ and K+ ion channels that maintain ionic homeostasis of Na^+^/K^+^ and expression levels of stress genes [81]. Similarly, Eida et al. [114] reported that endophytic strain treatment resulted in tissue-specific transcriptional changes of ion transporters and reduced Na^+^/K^+^ shoot ratios in *Arabidopsis* under salinity stress conditions.

According to the Habitat-Adapted Symbiosis Hypothesis, plants select endophytes from soils in order to increase tolerance to the specific stressors in that particular environment/habitat [115]. Endophytic colonization modulates gene expression levels to maintain stress tolerance. In a study, *Piriformospora indica* colonization into *Brassica campestris* subspecies *chinensis* confered salinity tolerance and higher expression levels of some specific salt tolerance genes, including SOS1 and SOS2, NHX-type [116]. The effective colonization of *Piriformospora indica* also elevated the antioxidant enzymes SOD; POD, CAT and elevated phytohormones, mainly SA, GA, that are directly involved in stress tolerance [117].

## 9. Studying the ‘Ome’ of Plant-Endophyte Interactions under Abiotic Stress

Endophytic microbes are known to modulate the genome, epigenome, proteome and metabolome of their hosts after inoculation to cope with abiotic stress. Plants with their modulated ‘ome’ after inoculation with endophytes bear better potential to ameliorate various abiotic stresses including drought and salinity. The molecular basis of endophytes in mitigating abiotic stress in crops is poorly understood. The recent developments in high-throughput technologies of sequencing and mass-spectroscopy based omics techniques have generated hopes for a detailed gene and protein study of molecular insights into the interaction of plant-endophytes during abiotic stress conditions (Figure 2).

The whole genome sequencing of endophytic bacteria revealed the presence of biofilm associated and fusaric acid resistant genes, which can play a crucial role in amelioration of abiotic stress in their hosts [118]. Similarly, genome-sequencing analysis of abiotic stress tolerant endophytic fungus *Pirifomospora indica*, showed the presence of stress tolerant genes [119]. Proteomic studies of the same fungus showed accumulation of photosynthesis, energy related proteins under drought conditions. Whole genome sequencing of endophytic fungi *Harpophora oryzae* and *Xylona heveae* demonstrated the presence of genes required for nutrient acquisition, which can provide abiotic stress tolerance to crops [120]. Many plant-symbiotic fungi, bacteria, yeasts and actinomycetes have been sequenced for their transcriptome, proteome and metabolome, and this has confirmed the presence of multiple plant growth promoting and stress tolerant traits [121].

Culture-independent sequencing approaches including metagenomics, metatranscriptomics and metaproteomics have emerged as new tools for studying the unexplored wealth of endophytes for conferring abiotic stress tolerance in plants. Shotgun metagenome analysis of uncultured microbe communities of endophytic bacteria revealed the population of Proteobacteria and Actinobacteria which can play a role in plant-growth promotion and abiotic stress tolerance [122]. Change in endophytic bacterial communities of wheat, as assessed by 16S rRNA sequencing, was associated with change in drought stress conditions [123].

Not only the endophytes, but the ‘ome’ of plants is also modulated during their interactions with endophytes while coping with abiotic stress. The ‘omics’ of endophytes also may be modulated by ‘horizontal gene transfer’ and synergism while interacting with their host crop [124]. Coutinho et al. [125] reported the influence of host crop *Oryza sativa* on gene expression of endophytic *Burkholderia kururiensis* M130 was related to biofilm regulation and iron transport. Some of the endophytic *Rhizobium* and *Xanthomonas* sp. associated with crops have shown transfer of genes responsible for plant adaptation and survival [126]. Comparative transcriptomics and proteomics studies associated with *Atractylodes lancea* in response to endophytic fungus *Gilmaniella* sp. AL12 revealed regulated plant metabolites, with upregulation in terpene skeleton biosynthesis and upregulated genes annotated as β-farnesene synthase and β-caryophyllene synthase [127]. Similarly, to understand the interaction of endophytic *Piriformospora indica* and host *Brassica napus*, an LC-MS/MS based label-free quantitative proteome technique was used, revealing the change in metabolic pathways, stress response and increase in stress adapting metabolites after endophytic interactions [128].

Understanding the roles of endophytes-plant interactions at a molecular level is crucial to understanding crop coping mechanisms to abiotic stress and may lead to more sustainable agriculture. The uncultured microbiome of endophytes can also be exploited for coping the abiotic stress using next generation of sequencing technologies.

## 10. Hurdles and the Way Forward

Externally applied endophytes have shown to be promising for amelioration of abiotic stress as evident by multiple studies; although in some cases their high performance does not always hold under field conditions. To ensure their efficiency in large scale application and commercialization of the endophyte-based products for amelioration of abiotic stress, several factors should be optimized as indicated below.

### 10.1. Lack of Standard Protocol for Surface Sterilization and Endophyte Isolation

Isolation of endophytes is a primary step toward developing applications in crops using endophytes. However, there is still a lack of consensus for standard surface-sterilization techniques to remove the epiphytic microbiota from the plant surface. Currently for plant surface sterilization, several disinfectants such as ethanol (70%) and bleach (5%, 3%, 2.7%) for different time intervals have been used, but sometimes higher concentrations of the sterilizing agents may damage the plant tissue, which affects the endophytic microbial community obtained [27]. In addition, the latest Next Generation Sequencing (NGS) improves our understanding of both epiphytic and endophytic microbiomes. However, experiments using non-cultivable microbes are difficult, and thus NGS still leaves limitations for practical application of endophytes [29,33].

### 10.2. Endophytes Should Ameliorate Multiple Abiotic Stresses and Should Be Good Plant Colonizers with Broad Host Ranges

Under laboratory conditions, endophytes can be screened for a single abiotic stress such as drought or salinity, but in field conditions, the host plant may face multiple stresses simultaneously. To cope with field stresses, the endophytes should confer tolerance to multiple abiotic stresses. Endophytic microbes should be able to colonize diverse plants and crops so that their application is not limited to few crops [129]. Moghaddam et al. [130] proposed the isolation of endophytes from extreme habitats (e.g., deserts, tundra, high elevations, etc.) for better amelioration of multiple stresses.

### 10.3. Endophytes Should Be Good Soil and Plant Competitors to Compete with Native Soil and Plant Microbes for Entry into Plant Tissues

One major impediment to use of endophytes under field conditions is the presence of native soil microbes and endophytes that outcompete the applied biostimulant microbes for entry into plants. Microbes that are poor competitors with other plant or soil microbes may be excluded from entry into plants because of the presence of other microbes that are in much higher concentrations in soils and in root tissues, effectively blocking their entry. For application under field conditions, biostimulant microbes should be better at entering into plant tissues than many other soil microbes. The qualities that make an endophytic microbe a better competitor are not yet fully understood.

### 10.4. Endophytes Should Not Be Plant or Animal Pathogens

Before commercialization, endophytes should be screened *in-planta* for pathogenicity or production of toxins. Some fungal and bacterial endophytes may not produce toxins in culture, but in plants or with other microbes they may produce toxic metabolites [131]. Many stress-tolerance-conferring endophytes, including *Colletotrichum* sp., *Alternaria* sp., *Fusarium* sp. and *Aspergillus* sp., may also be producers of mycotoxins [132].

### 10.5. Exogenously Applied Endophytes Should Not Interfere with Functions of the Plant Microbiome

For sustainable agriculture, it is necessary that applied endophytes, or the metabolites isolated from the endophytes, should not affect the host plant microbiome negatively. White et al. [30] showed that some endophytes enter into plant roots and interfere with the rhizophagy process and oxidative extraction of nutrients from native microbes in root cells. The interference with oxidative nutrient extraction from microbes in root cells was termed ‘endobiome interference’ [133]. Endobiome interference may occur if the microbe is highly resistant to reactive oxygen (superoxide) that is used in plant root cells to control and extract nutrients from internalized microbes [41]. Microbes that cause endobiome interference will enhance stress and reduce fitness in plants, causing growth inhibition and reducing nutrient absorption into plants. Incompatible endophytes may thus further hamper a plants’ capacity for stress tolerance [133,134]. The goal thus is to add endophytes that synchronize with the native microbiome, improve plant development, enhance nutrient acquisition and enhance the ability of plants to tolerate abiotic stresses [41]. It is imperative to check interactions between the native plant microbiome (particularly in roots) and the exogenously applied endophytes.

### 10.6. Endophyte-Based Formulations Should Be Optimized for Economical and Sustained Release under Field Conditions

For application of endophytes or endophyte-derived metabolites, they may be formulated in liquid or powder form. Under field conditions, activity of the biostimulant microbes or metabolites depend upon the type of formulation, humidity and temperature of the environment and the type of active ingredients [129]. For successful application of the formulations, it is necessary to optimize every parameter, specifically for abiotic stress conditions. Further, for a successful product, it should be economical. Economy is achieved when a single application of the microbe product results in persistence in the field. Multiple applications increase the cost of the biostimulant product. Several carriers such as chitosan, milk protein and maltodextrins have been used in formulations to increase shelf life and support initial inoculum growth after application to plants [135].

### 10.7. Better Public Awareness, Biological Product-Friendly Government Policies and Streamlined Registration Processes Are Needed

The consumers of an endophyte-based product are commercial growers, gardeners and homeowners. The uncertainty regarding an unknown biological product may discourage the use of new biological technologies including those based on endophytes. Articles for general audiences, awareness programs, workshops and outreach activities should be conducted at grass root levels to educate potential consumers and local vendors regarding endophyte modes of action and benefits to the environment and human health. Government policies and the registration process for agricultural biostimulants differ from country to country [136]. Governments should support endophyte-based biostimulants by changing policies and laws and allowing easier registrations of biological products. Government and industry partnerships to fund research on endophyte-based technologies could help move endophytes from the lab to the market. The results of this effort would be a less contaminated environment and a more sustainable agricultural system that has increased resilience to confront future climate perturbations.

## 11. Conclusions and Future Perspectives

Microbial endophyte biology is a growing field of research. The increasing output of research articles over the past two and half decades show that there has been an increasingly growing interest among researchers in the study of endophytic microbes. A significant body of knowledge has been accumulated over these years with regard to endophytic microbes and their effects on plants. We now know that communities of microbes [30] colonize plants in their shoots and roots. In many cases, microbes actually enter into plant cells themselves [30,41,137,138]. Studies on intracellular microbes involved in the rhizophagy cycle suggest that the interaction between endophytic microbe and the plant may be very intimate to the extent of a direct protoplast interaction within plant cells [30,41]. What is currently lacking is knowledge of the intimate microbe cell to plant cell interactions or ‘cross talk’ that results in all the beneficial effects in plants. What are the ‘words’ (or ‘signals’) uttered between endophytes and host plant cells that result in oxidative stress tolerance in plants? What is known is that plants respond to this interaction with intracellular endophytes by secretion of ROS [30,41]. The jury is not in yet, but it may be the host response to the endophytes with ROS (superoxide) that results in plant expression of increased oxidative stress coping systems. The signal sent to host cells that triggers the oxidative response may be the key to understanding the endophyte-host interaction. It is in that cross talk between endophyte and host that determines if the plant recognizes the microbes as friendly endophytic microbe or pathogen. We look toward the future when we may learn more about this intimate conversation between endophyte and host cells. Current and future research must focus on microbial endophytes to improve plant/crop productivity and create a more sustainable agricultural system where environmental degradation due to excessive agrochemicals is minimized.

## Figures and Tables

**Figure 1 microorganisms-09-01729-f001:**
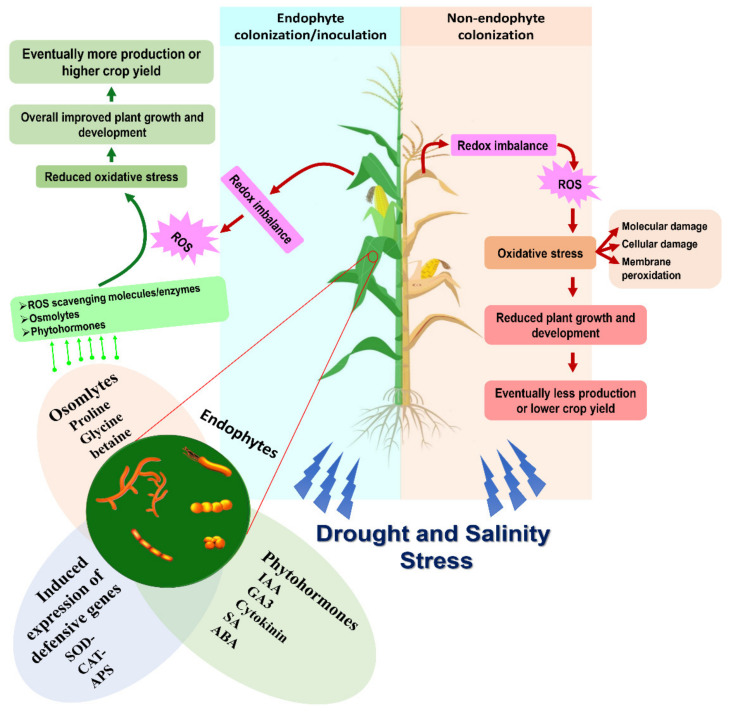
Overview of endophyte-mediated mechanisms for drought and salinity stress management in crop plants.

**Figure 2 microorganisms-09-01729-f002:**
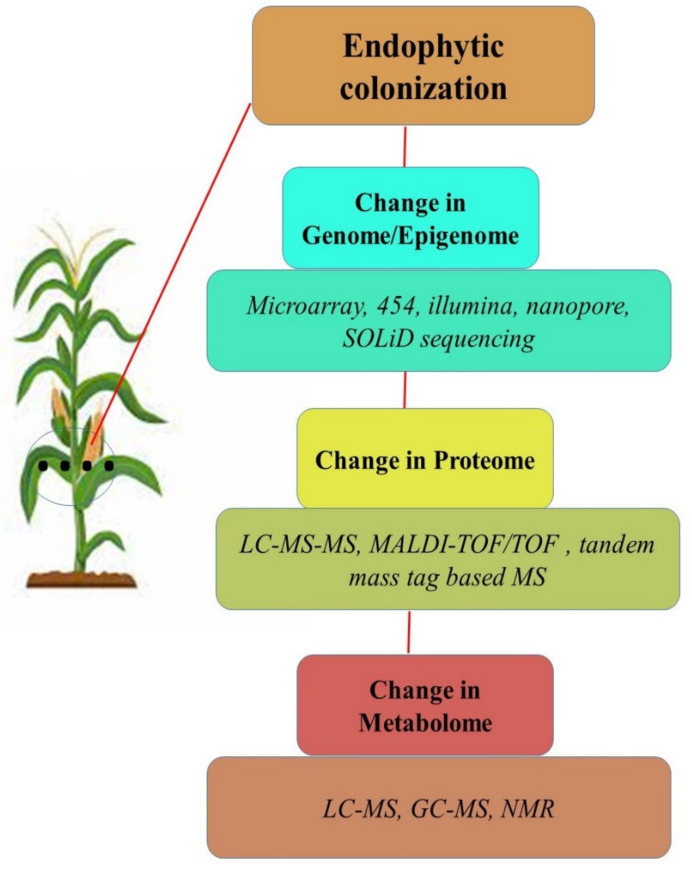
Overview of the ‘ome’ of plant-endophyte interactions under abiotic stress.

**Table 1 microorganisms-09-01729-t001:** Examples of studies reporting beneficial activities between microbial endophytes and their plant host under drought and salinity stress conditions.

Endophytic Strain	Type	Type of Stress	Mechanism of Stress Amelioration and/or Beneficial Activity	Plant Host	Ref.
*Phoma* species	Fungi	Drought	Increased Proline Peroxidase (POD), Catalase (CAT), Superoxide dismutase (SOD)	*Pinus tabulaeformis*	[66]
*Glomus mosseae*, *G*. *versiforme* and *G*. *diaphanum*	Fungi	Drought	Increment of peroxidase activity and beneficial effects on soil structure	*Poncirus trifoliata*	[67]
Endophyte consortia (*Rhodotorula graminis*, *Burkholderia vietnamiensis*, *Rhizobium tropici*, *Acinetobacter* *calcoaceticus*, *Rahnella* sp., *Burkholderia* sp., *Enterobacter asburiae*, *Sphingomonas yanoikuyae*, *Pseudomonas* sp., *Curtobacterium* sp.)	Fungi + bacteria	Drought	Reduced damage by reactive oxygen species (ROS), Increment of IAA	*Populus* sp.	[68]
*Bacillus*, *Achromobacter*, *Klebsiella* and *Citrobacter*	Bacteria	Drought	Production of 1-aminocyclopropane-1- carboxylate (ACC) deaminase	*Capsicum annuum* L.	[69]
*Burkholderia phytofirmans* PsJN and *Enterobacter* sp. FD17		Drought	Reduced H_2_O_2_ induced damage	*Zea mays* L.	[71]
*Streptomyces coelicolor* DE07, *S. olivaceus* DE10 and *Streptomyces geysiriensis* DE27	Bacteria	Drought	Phytohormone (IAA) synthesis and increment in water stress tolerance	*Triticum aestivum*	[72]
*Bacillus* sp. Acb9, *Providencia* sp. Acb11, *Staphylococcus* sp. Acb12, *Staphylococcus* sp. Acb13 and *Staphylococcus* sp. Acb14	Bacteria	Drought	Production of indole acetic acid, ACC deaminase and promotion of plant growth	*Ananas comosus, Vigna radiata*	[73]
*Pseudomonas* spp., *Acitenobacter brumalii* strain MZ30V92, *Enterobacter asburiae* strain MRC12, *Sinorhizobium meliloti* strain MRC31	Bacteria	Drought	Multiple plant growth-promoting traits	Not evaluated	[74]
*Pantoea alhagi* strain LTYR-11Z^T^	Bacteria	Drought	Increment on accumulation of soluble sugars, decreased accumulation of proline and malondialdehyde, and decreased degradation of chlorophyll in leaves of drought-stressed wheat plants	Arabidopsis and wheat	[75]
*Bacillus subtilis* B26	Bacteria	Drought	Upregulation of the drought-response genes, such as *DREB2B-like*, *DHN3-like* and *LEA-14-A-like* and modulation of the DNA methylation genes, such as *MET1B-like*, *CMT3-like* and *DRM2-like*, that regulate the process	*Brachypodium* *distachyon*	[76]
*Ampelomyces* sp. and *Penicillium* sp.	Fungi	Drought and salinity	Enhanced plant growth, stress tolerance, recovery and fruit yield	Tomato plants	[77]
*Bacillus subtilis BERA 71*	Bacteria	Salinity	Enhanced level of ROS scavenging antioxidant enzymes (superoxide dismutase, peroxidase, catalase)	*Cicer arietinum* seedling	[78]
*Streptomyces* sp.	Bacteria	Salinity	Increased proline, K+, Ca+ and water contents and decreased ethylene, ROS, Na+ and Na+/K+ ratio	*Oryza sativa* seedling	[79]
*Epichloë bromicola*	Fungi	Salinity	Increased photosynthesis, chlorophyll content, antioxidant capacity and glycine betaine content	*Hordeum* *brevisubulatum* Seedling	[80]
*Curvularia* sp.	Fungi	Salinity	Elevates antioxidant enzymes (SOD and APX)	Poplar plant	[81]
*Piriformospora indica*	Fungi	Salinity	Modulation of the expression levels of the major Na+ and K+ ion channels and balanced ion homeostasis of Na+/K+	*Arabidopsis thaliana*	[82]
*Brachybacterium paraconglomeratum*	Bacteria	Salinity	Enhanced level of proline, MDA, IAA in the inoculated plants	*Chlorophytum borivilianum*	[83]
*Trichoderma harzianum*	Fungi	Salinity	Reduces lipid peroxidation	*Lycopersicum* *esculentum* seed	[84]
*Piriformospora indica*	Fungi	Salinity	Enhanced plant growth and attenuated the NaCl-induced lipid peroxidation, metabolic heat efflux and fatty acid desaturation in leaves. In addition, significantly elevated the amount of ascorbic acid and increased the activities of antioxidant enzymes catalase, ascorbate peroxidase, dehydroascorbate reductase, monodehydroascorbate reductase and glutathione reductase	*Hordeaum Vulgare* *Seedling*	[85]

## Data Availability

Not applicable.

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
