# Peer review of "The Potential Application of Endophytes in Management of Stress from Drought and Salinity in Crop Plants"

_microorganisms, 2021, doi:10.3390/microorganisms9081729_

Round 1

Reviewer 1 Report

The current review focuses on the presence, diversity and role of endophytes in plant stress management. The article is of adequate length and it is divided into sections, including 8 that focus on specific topics regarding endophytes, stress factors and their management, signaling molecules, and omics studies.

Although the 8 sections defined topics that are of interest and well thought of, in reality I felt there was a lack of depth in terms of content. The majority of sections appeared to be a summary of examples and not a true discussion and review of the processes and mechanisms involved. A few examples:

1) in section 3, there is some repetitions throughout the text and there is a lack of detail in the description of the actual processes involved in the entry and colonization of hosts cells/plant organs by endophytes.

2) section 4 does not add a lot of value to the article as there are several questions that are not considered - are all ROS mitigated by the same enzymes? How does this vary? Are the same ROS molecules produced in response to different stress types? (the latter question is actually slightly considered in line 321)

3) section 5, only the text in lines 190-192 actually covers the topic described in the section title.

4) section 6 should review endophyte mediated abiotic stress management, but how is this management performed? What are the mechanisms/processes that the endophyte improves/modifies in the host? This section only gives examples of beneficial activities, which is probably not the most relevant information. It would probably be better to develop in more detail the information given in Table 1 (particularly in column #3). Regarding Table 1, there are no studies that have focused on other stress conditions rather than drought?

5) in section 7 (lines 258-261), the authors refer that certain phytohormones are commonly synthesized by endophytic microbial strains. It would be interesting to see some more information/discussion regarding if these molecules have any role/effect for the microbial strain itself, that is, why do they produce it? Is it in response to any signal from the plant/environment? Do they produce it because they also suffer due to the stress conditions? Or is it in response to the host's stress? An example of interesting information that I believe to be adequate to this kind of review is given in lines 291-297. (another good example is given in section 8, lines 363-365. These represent good examples since they actually explain mechanisms and processes).

6) in section 9, Figure 2 is not the most informative one - it does not really gives the reader information about plant-endophyte interactions under abiotic stress. It just depicts the normal process of changes at different levels and the techniques used to study each level. Again, this is an example of the lack of depth of information seen throughout the manuscript. From the figure legend I would expect to see detailed information of what mechanisms/processes are altered in plant-endophytes relationships during abiotic stress conditions in comparison with the "normal" not under stress conditions. Note: while reviewing bibliography for this section I came across some interesting studies that would be worth looking at, including the 2021 review article by Xu et al. entitled "Holo-omics for deciphering plant-microbiome interactions" or the Broberg et al. 2018 article "Integrated multi-omic analysis of host-microbiota interactions in acute oak decline". This is an indication that the authors should verify if any other relevant previous works are being missed.

Other comments include:

  • avoid repetition of keywords that are already in the article title.
  • review English and phrasing to avoid word/expression repetition. Examples: lines 40-42 repetition of "osmotic stress"; lines 48-49 repeat "affect growth and productivity of plants" as in line 47; lines 86-88 sentence starts with "Although" and it seems incomplete; lines 121-124 the idea is not clear; lines 156-167 there are a lot of repetitions of the word stress and the sentence construction is difficult to follow; lines 180-181 appear to repeat information given in section 4; line 230 the difference between "enhanced" and "promoted" drought tolerance is not clear. Furthermore, several sentences have misplaced commas that break the ideas in an unnatural way that makes comprehension difficult at times.
  • there are several specific genus/specific names missing the italics.
  • in section 2, lines 110-111, it is not clear how endophytes can help activate stress responsive/induced genes that are not usually activated under stress. Considering that they are stress responsive/induced it is to be expected that they would be activated on those conditions - so why are they not activated and how do the endophytes activate them?
  • Figure 1 is supposed to depict endophyte mediated mechanisms in response to drought and salinity stress. But in the figure itself, we can see that the authors have considered abiotic stress as a whole (including other stress factors). Possible questions regarding this figure are: does the microbiome impact the same independently of the stress factor (as shown in the figure) or are there different response types depending on the stress factor or even microorganisms present?
  • in line 127, it appears that the authors include biofilms as a secreted substance. But these are in fact microbial communities. Please review.
  • in line 133, the authors state "The nature of a microbial strain" - can the authors clarify? Is it the nature of the relationship between strain and host? Or do the authors wanted to refer to some intrinsic characteristic?
  • in line 326, the authors state "Endophytes induce synthesis of antioxidants" - clarify if the antioxidants are produced by the endophyte or by the host in response to the presence of the endophyte. In other words, explain the process by which endophytes induce such production.
  • in lines 403-406, how can those communities/microorganisms play a role in plant-growth promotion and stress tolerance? The metagenome analysis showed the community present, but how can the authors state which was their role? And in lines 406-407, the sentence is not clear - changes in bacterial community were detected by 16S and those changes were shown to be associated with stress response? How was that association made, that is, the new community detected under stress included more species of a certain group that has more beneficial activities or improves host fitness?

Author Response

Reviewer 1 comment

The current review focuses on the presence, diversity and role of endophytes in plant stress management. The article is of adequate length and it is divided into sections, including 8 that focus on specific topics regarding endophytes, stress factors and their management, signaling molecules, and omics studies.

Although the 8 sections defined topics that are of interest and well thought of, in reality I felt there was a lack of depth in terms of content. The majority of sections appeared to be a summary of examples and not a true discussion and review of the processes and mechanisms involved. A few examples:

  • in section 3, there is some repetitions throughout the text and there is a lack of detail in the description of the actual processes involved in the entry and colonization of hosts cells/plant organs by endophytes.

Response: Thanks for the comment. We have added additional details throughout this revision, especially regarding colonization of plants and needs going forward. 

  • section 4 does not add a lot of value to the article as there are several questions that are not considered - are all ROS mitigated by the same enzymes? How does this vary? Are the same ROS molecules produced in response to different stress types? (the latter question is actually slightly considered in line 321)

Response: ROS (reactive oxygen species) are of different types and they can vary  in response to various exogenous and indigenous factors.  Similarly the antioxidant enzymes are also of different types and they respond accordingly under different  stresses. The reviewer is correct.  We added a bit more, however, we have not wished to include too many details on ROS.  Another review could cover ROS  in more detail.

  • section 5, only the text in lines 190-192 actually covers the topic described in the section title.

 Response: We modified the section title and have expanded the coverage in this section.

4) section 6 should review endophyte mediated abiotic stress management, but how is this management performed? What are the mechanisms/processes that the endophyte improves/modifies in the host? This section only gives examples of beneficial activities, which is probably not the most relevant information. It would probably be better to develop in more detail the information given in Table 1 (particularly in column #3). Regarding Table 1, there are no studies that have focused on other stress conditions rather than drought?

 Response:  This review is focussed on drought and salinity stress, as we had also modified the title of the paper.  Table 1 is now extended and contain examples of salinity stress. As per concerns about mechanism and modification in the host by endophytes we have already explained in section 6,7 and 8.

5) in section 7 (lines 258-261), the authors refer that certain phytohormones are commonly synthesized by endophytic microbial strains. It would be interesting to see some more information/discussion regarding if these molecules have any role/effect for the microbial strain itself, that is, why do they produce it? Is it in response to any signal from the plant/environment? Do they produce it because they also suffer due to the stress conditions? Or is it in response to the host's stress? An example of interesting information that I believe to be adequate to this kind of review is given in lines 291-297. (another good example is given in section 8, lines 363-365. These represent good examples since they actually explain mechanisms and processes).

Response:  Thanks for the comments. We improved this section. However, because phytohormone modulation is also a broad topic. We are planning to review this theme also in a separate paper. In this review we had already covered the aspect of phytohormone modulation in salinity and draught stress management.

6) in section 9, Figure 2 is not the most informative one - it does not really gives the reader information about plant-endophyte interactions under abiotic stress. It just depicts the normal process of changes at different levels and the techniques used to study each level. Again, this is an example of the lack of depth of information seen throughout the manuscript. From the figure legend I would expect to see detailed information of what mechanisms/processes are altered in plant-endophytes relationships during abiotic stress conditions in comparison with the "normal" not under stress conditions. Note: while reviewing bibliography for this section I came across some interesting studies that would be worth looking at, including the 2021 review article by Xu et al. entitled "Holo-omics for deciphering plant-microbiome interactions" or the Broberg et al. 2018 article "Integrated multi-omic analysis of host-microbiota interactions in acute oak decline". This is an indication that the authors should verify if any other relevant previous works are being missed.

 Response:   Thanks for the comment. We have examined the references you indicated. Our intent with this section was to provide an overview of the utility of omics approaches to developing a better understanding of mechanisms of endophyte-enhanced stress tolerance. We have not intended to do a comprehensive coverage.

Other comments include:

  • avoid repetition of keywords that are already in the article title.

Response:  done

  • review English and phrasing to avoid word/expression repetition. Examples: lines 40-42 repetition of "osmotic stress"; lines 48-49 repeat "affect growth and productivity of plants" as in line 47; lines 86-88 sentence starts with "Although" and it seems incomplete; lines 121-124 the idea is not clear; lines 156-167 there are a lot of repetitions of the word stress and the sentence construction is difficult to follow; lines 180-181 appear to repeat information given in section 4; line 230 the difference between "enhanced" and "promoted" drought tolerance is not clear. Furthermore, several sentences have misplaced commas that break the ideas in an unnatural way that makes comprehension difficult at times.

Response: Done

  • there are several specific genus/specific names missing the italics.

Response: Done

  • in section 2, lines 110-111, it is not clear how endophytes can help activate stress responsive/induced genes that are not usually activated under stress. Considering that they are stress responsive/induced it is to be expected that they would be activated on those conditions - so why are they not activated and how do the endophytes activate them?

Response: With endophyte infection, various stress response traits, including ROS generation, are activated, causing plants to respond with antioxidants which leads to increased oxidative stress resistance in plants.  In addition, endophytes generally produce antioxidative compounds such as nitrogen oxide that directly reduces ros and stress. We have tried to explain this in the text.

Figure 1 is supposed to depict endophyte mediated mechanisms in response to drought and salinity stress. But in the figure itself, we can see that the authors have considered abiotic stress as a whole (including other stress factors). Possible questions regarding this figure are: does the microbiome impact the same independently of the stress factor (as shown in the figure) or are there different response types depending on the stress factor or even microorganisms present?.

Response: The figure represents overall factors associated with abiotic stress and endophyte mediated stress management. Now we had already modified the Fig.1 for better presentation.

  • in line 127, it appears that the authors include biofilms as a secreted substance. But these are in fact microbial communities. Please review.

Response: We deleted the term Biofilm

  • in line 133, the authors state "The nature of a microbial strain" - can the authors clarify? Is it the nature of the relationship between strain and host? Or do the authors wanted to refer to some intrinsic characteristic?

Response: Here the nature of strain means   mutualistic, pathogenic and non-pathogenic 

  • in line 326, the authors state "Endophytes induce synthesis of antioxidants" - clarify if the antioxidants are produced by the endophyte or by the host in response to the presence of the endophyte. In other words, explain the process by which endophytes induce such production.

Response: Thanks for comments. Actually, endophytes induce synthesis of antioxidants by plants—but they also produce antioxidants [see reference 41]. We have tried to clarify this in the text.  

in lines 403-406, how can those communities/microorganisms play a role in plant-growth promotion and stress tolerance? The metagenome analysis showed the community present, but how can the authors state which was their role? And in lines 406-407, the sentence is not clear - changes in bacterial community were detected by 16S and those changes were shown to be associated with stress response? How was that association made, that is, the new community detected under stress included more species of a certain group that has more beneficial activities or improves host fitness?

Response:  Thanks  for this comments.  We modified this paragraph accordingly

Reviewer 2 Report

The manuscript, is a review which delas with an overview of endophytes action and potential help to manage abiotic stress.

The topic is of great interest and the idea of the manuscript is really of importance for readers and users of this type of article in research and teaching.

Nevertheless, many shortcomings limit the scope of the manuscript.

The first one is the real originality of this manuscript. Please see this review.

https://doi.org/10.3390/pathogens10050570

Moreover, authors display an attractive and tempting title concerning "plant abiotic stress management", but the illustrations and especially Table 1, which gives an overview of the situation, only present examples of drought.  There are many studies on abiotic stresses and the impact of endophytes that have not been valued and/or not cited by the authors.

https://doi.org/10.3390/metabo11070428

https://doi.org/10.3390/microorganisms9071373 

https://doi.org/10.3390/agronomy11061167

https://doi.org/10.3390/antiox10060880

https://doi.org/10.3390/microorganisms9051050 

https://doi.org/10.3390/biology10050409 

 https://doi.org/10.3390/su13084422

The other critical point is the English language. Authors should get help with the editing of the English language. For example, and symptomatic of the whole manuscript, the title. There is not one abiotic stress, there are many.

Author Response

Response: Thanks for the comments.  We had thoroughly revised the MS and hope this revised MS will be better. We also revised the writing carefully. 

Reviewer 3 Report

The review does not look to be fresh and advanced. Many references are reviews too. A few of real experimental examples. A lot of confusions with mycorrhizal, soil, and antagonistic fungi, PGPR bacteria. Many words on the management of the plant productivity without examples of the real management. Conclusions seems are not novel and represents just general words. Some marks and comments in the file attached.

Author Response

Response: Thanks for the comments. However we had modified the MS  as per suggestion and also added a new section  to increase novelty of the paper.

Reviewer 4 Report

Article is interesting, but are some points for correction:

  1. In the Table 1 should be added compounds produced by endophytes, which have action on stress amelioration.
  2. In Table 1 "Lolium perenne" is not endophyte.
  3. Figure 1 in poor and unclear.
  4. Authors must correct names of organisms, and all should be written italics.

Author Response

Reviewer 4

Article is interesting, but are some points for correction:

  1. In the Table 1 should be added compounds produced by endophytes, which have action on stress amelioration.

Response: Table is now modified

  1. In Table 1 "Lolium perenne" is not endophyte.

Response:  deleted

  1. Figure 1 in poor and unclear.

Response: corrected

  1. Authors must correct names of organisms, and all should be written italics.

Response: corrected/ modified

Reviewer 5 Report

Reviewed paper concerns very interesting and topical issue connected with the possibility of abiotic stress relieve of plants through the use of endophytes. It was shown in it many interesting responses of plants on the action of endophyte microorganisms. Especially important are these concerning drought and soil salinity. In that manuscript were cited 126 papers, mainly indicted on favorable effect of endophytes on abiotic stress decrease of plants. The main part of the manuscript consist of text and 2 figures ( Figure 1. Overview of endophytic mediated mechanism of draught and salinity stress management in crop plants; Figure 2. Overview of the ‘ome’ of plant-endophyte interactions under abiotic stress). In my opinion at discussing issues Authors should present also documental materials (figures, tables) what would make that papers would be more varied and interesting. Moreover, in that paper this issue is presented more optimistic and there is lack of critical view of an elevated issue. There is a question 1) why after the many years of studies connected with endophytes these microorganisms are not use in plant production on the wider scale? and 2)how is the efficiency of the preparations with endophytes in limited of the abiotic stress, like for instance drought stress? Next, in the paper is lack of a section “ Problems connected with endophyte use as the preparations decreasing of abiotic stress of crops”.

Author Response

Reviewer 5 comments

Reviewed paper concerns very interesting and topical issue connected with the possibility of abiotic stress relieve of plants through the use of endophytes. It was shown in it many interesting responses of plants on the action of endophyte microorganisms. Especially important are these concerning drought and soil salinity. In that manuscript were cited 126 papers, mainly indicted on favourable effect of endophytes on abiotic stress decrease of plants. The main part of the manuscript consist of text and 2 figures (Figure 1. Overview of endophytic mediated mechanism of draught and salinity stress management in crop plants; Figure 2. Overview of the ‘ome’ of plant-endophyte interactions under abiotic stress).

 In my opinion at discussing issues Authors should present also documental materials (figures, tables) what would make that papers would be more varied and interesting. Moreover, in that paper this issue is presented more optimistic and there is lack of critical view of an elevated issue. There is a question 1) why after the many years of studies connected with endophytes these microorganisms are not use in plant production on the wider scale? and 2)how is the efficiency of the preparations with endophytes in limited of the abiotic stress, like for instance drought stress? Next, in the paper is lack of a section “ Problems connected with endophyte use as the preparations decreasing of abiotic stress of crops”.

Reviewer 5 response

  • why after the many years of studies connected with endophytes these microorganisms are not use in plant production on the wider scale?

Response: Endophytes have shown potential application in abiotic stress amelioration and they have been used in the field predominantly as biostimulant microbes. Many of these biostimulant microbes have not even been recognized as endophytes. 

In the journey of the endophyte from lab to land, they have to face many hurdles related to toxicity, safety of plant consumptions treated with endophytes, environment aspects and policies of different governing bodies., which further restricts their application on the wider scale. However, overcoming these limitations can result in the successful application of endophytes resulting in abiotic stress tolerance in field conditions in an economical and eco-friendly manner.

There are some examples which show the effective application of endophytes on larger scales. Kauppinen et al., 2016 discussed the role of endophytes in increased productivity of forest grasses. Johnson et al., 2013 estimated that endophytes mediated improvement in plants resulted in contribution of $200 million per annum to New Zealand economy.

If the hurdles related to commercialization of the endophytes would be removed, it definitely bears the capacity to bring a change in abiotic stress management and sustainable agriculture.

As per the reviewer’s suggestion, a subsection ‘hurdles and way forward’ has been incorporated to discuss problems and solutions.

 2)how is the efficiency of the preparations with endophytes in limited of the abiotic stress, like for instance drought stress?

Response : Authors are thankful to the reviewer for this critical comment. The efficiency of the preparation depends upon multiple factors such as type of preparation (powder, liquid, suspension), the active principle of the formulation (endophyte, crude extract, purified metabolite), storage conditions (temperature, humidity) and field conditions.

In lab, under ideal conditions, the efficiency of the endophytes in ameliorating abiotic stress is quite high, which has to be optimized further for their large scale application in field conditions.

A subsection ‘hurdles and way forward’ has been incorporated to discuss aspects asked by the reviewers in detail.   

3)Next, in the paper is lack of a section “ Problems connected with endophyte use as the preparations decreasing of abiotic stress of crops”.

Response: As per the reviewer’s suggestion, a subsection ‘hurdles and way forward’ has been incorporated to discuss problems to be overcome to better use endophytes in agriculture.

Round 2

Reviewer 2 Report

The manuscript was greatly improved by considering the reviewers' recommendations. It is now clearer and focused only on drought and salinity.

More  examples were added in table 1.

I also appreciate the item 10 which was added. Nevertheless, some points were lacking among them isolation and production of microorganisms (the first step) please add this point.

Author Response

Thanks for the comment. We have added a section of “Lack of standard protocol for surface sterilization and endophyte isolation” in the section 10.a (Highlighted in RED) as per suggestion and modify the table for better presentation

Reviewer 3 Report

The authors improved the text of the review. However, there are still some gaps for a polishing. 1) Confusions among fungi and bacteria (see table 1). Are penetration paths, mode of action, application methods similar for pro- and eukaryotes ? 2) Please, provide more experimental examples and data on effective concentrations for protective effects of particular endophytes from different groups.  

Author Response

Thanks for the comment. We have added a separate column to indicate fungal and bacterial strains in table. Additionally we have also added the experimental details in the table. Regarding concern about penetration path, mode of action, application methods of pro and eukaryotes “would say that the penetration paths, application methods and mechanisms of action are similar for eukaryotic and prokaryotic microbes.  Both tend to enter into plant tissues at meristems and may even become intracellular in meristematic cells--particularly in roots.  One would apply them in a similar way on seeds--and modes of action are similar--with one difference--there is no nitrogen fixation with fungal endophytes.  Otherwise nutrient acquisition is similar.  

Reviewer 4 Report

Authors corrected some points. However, in Table 1 are not added compounds produced by endophytes, which have action on stress amelioration. Authors describe only type of action decrease of ROS or changes in genes expression. In Table are not included names of bacterial/fungal compounds, which lead to these effects.

Author Response

Thanks for the comment. We have added a separate column to indicate fungal and bacterial strains in table. Additionally we have also added the experimental details in the table (highlighted in red).

Reviewer 5 Report

I would like to inform you that I accept the changes which are made in the improved manuscript. My suggestion are used in the text so in my opinion that paper is more complex and informative. On the present version that manuscript may be published in Microorganisms.

Author Response

Thanks for this nice comment

Round 3

Reviewer 3 Report

I thank authors for the reply but disagree with them who did not give any example both in their remarks and in the section 3 (moreover, no Latin name of any microbe can be found in sections 1, 3-6 of the manuscript). It is unbelievable (especially, without any evidence) that bacteria and fungi from so different taxonomy groups have the same entry paths and mode of action. Just see the conclusion “We now know that communities of microbes [30] colonize plants in their shoots and roots. In many cases, microbes actually enter into plant cells themselves [30,41,137,138]. Studies on intracellular microbes involved in the rhizophagy cycle suggest that the interaction between endophytic microbe and the plant may be very intimate to the extent of a direct protoplast interaction within plant cells [30,41].” In my comments I proposed to give (or discuss) additional info on the concentrations of endophytes that favorable stress tolerance because it is clear that this effect is concentration depended. Well-established model system only allows to study plant-microbe interactions, but I did not see anything like this in the text. The visible disadvantage of the manuscript is no differentiation between bacteria and fungi. At least in the table they could be divided. The manuscript looks finalized for authors and it is up to them and Editors to publish a bit raw manuscript.

Author Response

Reply:  Thank you for your thoughtfulness. It does seem strange, but entry point for both fungi and bacteria tend to be into meristematic tissues.  This is where plant cells are releasing nutrient and where tissues are not covered by a waxy cuticle.  It is true that some microbes may remain intercellular while other become intracellular. Nevertheless, the entry points are the same.  Even the fungal Epichloe endophytes enter tissues in meristematic tissues.  The mode of entry seems determined more by the vulnerability of the plant meristematic tissues rather than to some special capabilities of the microbes. 

In terms of modes of actions.  These are still not fully understood.  Stress tolerance is a common theme with many fungal and bacterial endophytes.  There are hypotheses about how microbes trigger or induce stress tolerance. One is the ACC deaminase idea.  We have discussed it—but this really only applies to some bacterial endophytes—but may endophytes trigger oxidative stress tolerance whithout ACC deaminase. Another, mechanism is likely responsible for oxidative stress tolerance induced by most fungal and bacterial endophytes.  This is that most endophytes appear to trigger plant cells to increase secretion of reactive oxygen species—and this requires host plants to increase production of antioxidants—and this makes plants more resistant to oxidative stress. It is likely that these oxidative interactions between endophyte and host are responsible for most of the abiotic stress tolerances.  Of course, plants make other adjustments in coping with stress. But the primary mechanism is this oxidative tolerance adjustment that hosts make. 

However we have added two subsection in the PAPER as per your suggestion, highlighted in the red

Reviewer 4 Report

The authors did not understand the word COMPOUNDS? Compounds means chemicals, substances, metabolites. Not fungal and bacterial strains!!! I asked for adding in the Table COMPOUNDS PRODUCED BY ENDOPHYTES, which have action on stress amelioration. Compounds like gliovicin, peniprequinolene, gliovicinacetate, cytochalasin, pyrrocidine and others.

Author Response

We don’t  agree with reviewer suggestion. Reviewer suggest us to add the compound name in the table, when the author (cited in table) did not mentioned the name of compounds in his paper, then how we can add.  Secondly, we have already mentioned the action mechanism of endophytic microorganism through which they ameliorate the stress condition in the table.

I am mentioning here details about paper cited in table (reference 67-72)

67- Author not confirmed any compounds and used the term "may " for glomalin

68-  Paper discuss about genes, not mentioned any compounds

69 – no any compounds name mentioned

71- not mentioned any compounds

72- not mentioned any compounds

73 -not mentioned any compounds and so on..